# Developing Multimodal Narrative Genres in Childhood: An Analysis of Pupils' Written Texts Based on Systemic Functional Linguistics Theory

**Alejandra Pacheco-Costa [1,]** and **Fernando Guzmán-Simón [2]**

1　Department of Arts Education, Facultad de Ciencias de la Educación, Universidad de Sevilla,
　41013 Sevilla, Spain
2　Department of Language Education, Facultad de Ciencias de la Educación, Universidad de Sevilla,
　41013 Sevilla, Spain; fernandoguzman@us.es
*　Correspondence: apacheco@us.es

**Abstract:** Social sustainability embraces literacy development as a means by which children integrate their knowledge in society and become powerful and meaningful. In this context, the development of writing among young children requires the design of new teaching strategies that allow for the multimodal repertoire brought by children into the classroom. Systemic Functional Linguistics offers tools for the analysis of children's multimodal writing, which plays an important role in their literacy development. Our research was carried out in an urban context, the participants being 12 children aged 7 to 8. Data were collected through participant observation, conversations, and the analysis of documents and products generated by the children. From them, we analysed two stories written by two girls, which showed the way in which the children created meaning by combining verbal and visual modes, and how these modes interact (intersemiosis). The performance of a literacy task in which children are able to integrate their knowledge and heritage into the classroom, may constitute an interdisciplinary tool for their participation and engagement in the school, thus leading to a more equal society. In consequence, we propose that the integration of a genre-based pedagogy in the classroom should include greater awareness in teachers of the value of pupils' multimodal assessments.

**Keywords:** primary education; social sustainability; literacy; writing; genre; multimodal text

## 1. Introduction

Social sustainability has been described as the sum of formal and informal processes, systems and relationships that work together building healthy and liveable communities [1]. The non-physical factors supporting social sustainability include, among others, education, social capital and cultural tradition [2]. Educational as well as linguistic concerns influence also the achievement of equity, considered as one of the key elements of sustainable societies [2], both within and between generations [1].

A broad understanding of social sustainability supports literacy, as well as numeracy, music or movement, as part of the knowledge acquired by children in their communities, and brought by them into the classroom [3]. Following the interdisciplinary concept of social sustainability developed by McKenzie [1], and its materialisation in schools [4], we focus on young children's creative writing, considering it as a field for participation, intergenerational understanding and identity development.

Recent research on written communication during early childhood [5–9] has underlined the variety of multimodal texts surrounding children, and the way in which they develop a multimodal literacy [10] during their literacy development, both in and out of school [11–13]. Therefore, rather than being instructed in the understanding and creation of multimodal genres

in the classroom, children's acquisition of multimodality comes as part of the global process of their literacy development, which takes place in diverse settings. The current paper addresses the analysis of children's products in school from a functional and systemic approach to written communication. Defined in work by Michael Halliday [14], Halliday and Hasan [15] and other researchers [16–18] among others, Systemic Functional Linguistics (hereafter SFL) has been described as a systemic approach embracing a semantic-functional perspective of language, where the cultural and social context acquires a fundamental role in the analysis of the artefacts produced by children. This approach enables us to consider the way in which a specific individual uses and structures discourse in different contexts. At the same time, our research aims to present multimodal literacy development in the school as a tool for developing cultural relations between the school and the children's environment, considered as an indicator of social sustainability [1].

## 1.1. From a Literacy Approach to a Literacy Education

The SFL enables the analysis of literacy, understood as a social practice, applied to diverse genres within a specific context. The analysis of context, as proposed by Halliday and Hasan [15], integrates a complex perspective of communication. This perspective was on the basis of the Social Semiotics [19] and enabled a widening of the boundaries of the discourse analysis, as pointed by van Leeuwen [20] and Kress [21]. The SFL proposes a reformulation of the way in which literacy research approaches the discourse analysis in all its complexity, both verbal and visual [22].

The SFL approaches the study of language from its organisation and from the social context in which it is used with a specific communicative aim [23]. The analysis of texts according to the SFL emanates from a dual concept of language, that being functional and systemic [16]. From its functional perspective, language has always a communicative aim and, consequently, every text has a functional orientation within a particular context. In its systemic perspective, texts possess a precise structure in their different stages (phonology/graphology, lexicogrammar, discourse semantics), configuring a semiotic system in which the different meanings of the text are contained [23].

The current concept of literacy demands a deep shift both in the learning skill-based methods, unconnected to the children's heterogeneous contexts, and in the literacy mediator/expert/teacher role. The "genre-based approach" brings together not only the individual literacy practices and certain codification skills, but also communication as the main purpose of writing. Therefore, a specific genre hosts a purpose, stages, and a "register" that should be adapted to the context of the situation and to the communicative purpose of the speaker/writer [22]. SFL presents the text as a unit within the context of the situation. As Butt et al. state, text is "a purposeful, harmonious collection of meanings relevant in the context, unified by texture, or the way the meanings fit together in the text, and structure, or the organisation of the elements of the text" [24] (p. 22). The genre-based approach permits the analysis of the context from a sociocultural perspective, where texts have a specific and precise communicative aim, which determines the use of language [16].

Whereas genre is related to the context of culture, register ascribes to the context of situation. Register is defined by the analysis of the linguistic variation and by the context of situation—*where* and *when*—the communication takes place. From a sociolinguistic point of view, the linguistic variation of the text is defined by three elements: field, tenor and mode. These elements constitute the context of situation and conform to the three types of meaning that constitute a text: ideational (field), interpersonal (tenor) and textual (mode) [16] (see Figure 1).

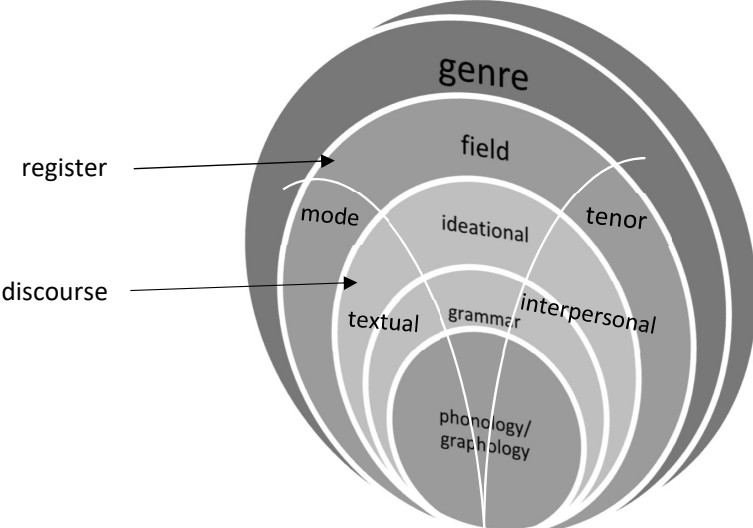

**Figure 1.** Genre, register and language in SFL (Systemic Functional Linguistics). Source: Rose and Martin [25] (p. 311).

The sociolinguistic approach of SFL describes the elements essential to characterise and predict the various linguistic registers: the field, the tenor, and the mode (Table 1). The theory of register depends on the situation in which the communication develops [16]. The analysis of these registers is a key aspect of examining a text. The register is linked to the linguistic variation in the text, when considered within the concrete situation of the writer [23]. Ultimately, the use of grammar contributes to the meaning-making provided by the metafunctions, through the analysis of transitivity, mood and theme of the text [26].

**Table 1.** Relations between language, registers and meanings.

| Registers | Language | Meanings | Description |
|---|---|---|---|
| **Field** | Ideational metafunction | Experiential | Express the experience of the world through language (transitivity system). |
| **Tenor** | Interpersonal metafunction | Interpersonal | Express the nature of the relation between the author and the reader, and the attitude of the author towards the theme exposed (mood types, system of polarity and system of modality). |
| **Mode** | Textual metafunction | Textual | Determines the way in which a text—oral or written—is organised, its relation with previous texts and the surrounding context (thematic organization). |

Sources: Halliday [19] and Eggins [16].

Our genre-based research adopts the SFL perspective of the Sydney School [25]. This approach considers that genres emanate from the social practices in which oral and written communication is developed. Martin states that "genre is concerned with systems of social processes, where the principles for relating social processes to each other have to do with texture—the ways in which field, mode and tenor variables are phased together in a text" [23] (p. 12). Genre as a social practice presents a structure easily recognisable by the recipients of the text. This structure is built in phases depending on the genre [27]. Therefore, genres become textual recurrent models, identifiable within a context of culture. The configuration of genres from the SFL perspective enables the construction of a text drawing on a textual model, and eases the recognition of the genre by its recipient [16].

Martin and Rose [28] pose two criteria for the categorisation of genres. The first criterion is the presence or absence of certain elements, such as time, process explanation, description of objects, or the number of points of view. The second criterion is the organisation of sequences according to the genre. As stated by Rose, story genres can be distinguished according to the presence or absence of sequence in time (news reports vs. other stories) and the presence or absence of a complicating event (recount vs.

narrative); factual genres, according to whether they explain processes or describe things (explanation vs. report); argument genres according to whether they argue for a point of view or discuss two or more points of view (exposition vs. discussion). Secondly, the organization of each genre can be distinguished by recurrent local patterns, such as the narrative stages OrientationˆComplicationˆResolution, or the exposition stages ThesisˆArgumentsˆReiteration [27] (p. 209).

Our analysis is focused on written stories, more specifically on narratives, where the story resolves the complications [28]. In this sense, the narrative genre is conformed by three stages—OrientationˆComplicationˆResolution—with a clear temporal sequence.

## 1.2. Multimodal Texts

The educational perspective of social sustainability begins with the immersion of children into their world, and involves literacy, arts, and language [29]. Schools should be able to provide spaces fostering equity through participation [2], and incorporating the children's voice. The relevance of multimodal writing in early childhood literacy finds its place among the children's written products, and needs to be analysed accordingly. The recent works by Bateman, Wildfeuer, and Hiippala [30], Jewitt, Bezemer, and O'Halloran [31] and O'Halloran, Tan, and Wignell [32] have posed a deep revision of the theoretical framework that connects SFL and Multimodal Discourse Analysis. Together, they have enabled an approach that combines SFL [28,33] and the visual discourse of "reading images" [34,35]. This theoretical framework has allowed some of the most representative analyses of multimodal children's texts in recent years, such as those carried out by Painter [36], Moya Guijarro [37] and Painter, Martin, and Unsworth [38]. For instance, Painter, Martin, and Unsworth [38] have described in detail the ways in which the verbal and visual modes interact in a multimodal text, producing integrated or complementary layouts, or the nature of actions depicted in the speech bubbles, among other aspects.

Nowadays, the process of teaching and learning to write requires the integration of new educational programmes, where the children assume a new role within the communicative setting of the school [39]. From this perspective, the different contexts with which children interact, in and out of school, are present in their written texts. In them they interact with diverse semiotic systems right from the first written products of their initial literacy [6]. Therefore, the construction of written multimodal texts has acquired a relevant role in initial literacy in school, as soon as it adopts "a set of systems of meaning, all of which interrelate" [15] (p. 4).

The development of media and information, and communication technology has widened the repertoire of multimodal genres that children encounter [21]. In such a context, texts are "co-deployed across various modalities ( . . . ) as well as their interaction and integration in constructing a coherent text" [40] (p. 74). In our research, based on the analysis of children's written products, multimodal texts are defined as the construction of meaning, and as the result of the interaction between verbal and visual semiotic codes [41]. Multimodal texts are heterogeneous, and modify both the writing practices of children and the process of decoding the diverse logonomic systems [35] of the written text [42].

Understanding a multimodal text requires the "resource integration principle" [43], the theorisation of "intersemiosis" [44] and "resemiotization" [45]. The "resource integration principle" relies on the construction/group of senses coming from the diverse semiotic sources displayed in the written text. The interaction between different semiotic sources (also named "intersemiosis") implies a re-negotiation of the meaning of each one of these sources within a specific context. Following O'Halloran, intersemiosis may be defined as "a convergence of meaning (co-contextualizing relations of parallelism) and/or divergence of meaning (re-contextualizing relations of dissonance) in multimodal texts" [44]. Besides, this analysis of the multimodal text focuses on the multi-semiotic nature of representation, but also "seeks to underscore the material and historicized dimensions of representation" [45]. As pointed by Iedema [45], this process of negotiation of the meaning is called "resemiotization" and may be applied to the semantic-discursive dimension of the source, as well as to its material logic.

Likewise, the characteristics of the multimodal text are configured by the social function of the semiotic codes used in the written communication. These characteristics are structured according

to three metafunctions [19], which become an essential tool to understand a multimodal text [38]. The metafunctional principles embed the functionalities and the structure of the semiotic sources, as well as their interaction and integration with other semiotic codes within a multimodal text [31]. The recurrent use of a source and a mode in a context, with a particular communicative purpose, enables the development of its *potential meaning* [20] or affordance [46]. Thus, the metafunctions determine both the use and understanding of the diverse semiotic sources and modes, and the intersemiotic relation between the meanings of the text's different sources. Martinec and Salway [47] suggested a categorisation where the intersemiosis of visual and verbal modes is based on the metafunctions and their dependency relationship. In short, O'Halloran et al. have posited that "( . . . ) the effect of multimodal forms of semiosis on meaning-making is multiplicative and multidirectional" [31] (p. 440). The differences between the visual text (design, structure, colour, etc.) and the verbal text (sequence of events, temporal expression, grammar, etc.) permit the configuration of complex meanings through the processes of intersemiosis and resemiotization [41]. These meanings depend on the social use of the children's written products. An approach to these multimodal texts in childhood has been one of the foci of the current research.

## 2. Materials and Methods

### 2.1. Our Research

Our research presents the results of an intervention carried out following a genre-based model [24], for which the genre-based pedagogy draws on a concept of genre described as "concerned with how a culture maps ideational, interpersonal and textual meaning onto one another in phases as a text unfolds" [48] (p. 55). Along with these concepts inherited from SFL as defined by Halliday [19], we have also drawn on the methodological contribution of Bernstein [49] and Bruner [50]. All these features contribute to a genre-based pedagogy, which focuses on an explicit learning of language, always relying on the ways language is used in society [48]. The learning of metalanguage based in genres, as proposed by Rose and Martin [24], assumes a fundamental role in school education, insofar as its aim is to make children aware of language systems from an early age.

To this end, authors such as Hyland [51], Kalantzis and Cope [12], Rose and Martin [24], and de Silva Joyce and Feez [22], have proposed complementary methodologies. Genre-based pedagogy is based on scaffolding oriented to the children's literacy development. This approach to literacy based on genres merges both oral and written language, and manifests as a systematic and explicit training in the school. All the tasks are oriented towards the development of this metalanguage, which emanates from the concept of literacy as a social practice, with the teacher assuming the role of expert [22]. According to Hyland [51], this role includes the use of scaffolding through guide questions, instructions or templates [52], which have a collaborative function fundamental to the children's learning, leading to individual development of metalanguage. The tasks are presented as challenges where the children receive scaffolding depending on their level of competence, and where they undertake a learning process integrated by three stages: (1). Deconstruction stage; (2). Joint construction stage; and (3). Individual construction stage [24].

In this context, the genres of multimodal texts pose a new challenge for literacy education in the 21st century. Multimodal texts require teaching staff and children to rethink how to approach the making of meaning. For this reason, it is also necessary to assume the importance of multimodality in the shaping of genres [53] and to develop a metalanguage able to facilitate the writing of text in all its complexity [40].

Our genre-based intervention took place in a primary school classroom of children aged 7 to 8 years old and was directed towards the improvement of reading and writing skills within the narrative genre. It consisted of a weekly session lasting one hour and a half, from mid-October 2019 to mid-March 2020. The intervention followed the stages described by Rose and Martin [24]:

- Deconstruction stage: In this stage, children became aware of the multiple narratives surrounding them, and were able to notice and reflect on their components and function. As the children were

still not confident writers, this stage was developed mostly orally. They also incorporated their own narrative repertoire, based on traditional tales, popular culture and audio-visual media.

- Joint construction stage: This stage was usually designed as a game—guessing games, role play, cops and robbers, etc.—in which the children, organised into groups, were asked to perform a task related to the deconstruction stage. All the tasks contained a multimodal approach.
- Individual construction stage: In this stage, the children had to carry out a task similar to the one of the previous stage, in pairs or individually. They were given some freedom, so they were permitted to make choices and connect the task with their daily life and experience. Frequently, the products generated were displayed or read aloud in the classroom.

The scaffolding strategies offered to the children during the intervention relied on their existing, intuitive knowledge of the narrative genre, acquired through oral and written texts, and through various media. From these sources, the children could become aware of the metalanguage of the narrative texts. The visual elements surrounding them, linked to the specific narrative genres, were also part of the scaffolding.

In order to provide a deeper understanding of the metalanguage of narrative texts, the sessions focused on precise topics, such as the narrative or descriptions of people or the situation. A brief synthesis of the whole intervention may be found in Table 2.

**Table 2.** Genres undertaken with children in the classroom.

| Genre | Sessions | Products Generated |
|---|---|---|
| Narrative | Four | Written multimodal stories about Halloween. Written story narrating the sequence of a set of drawings. Written multimodal narration of a movie or episode in a three-stage sequence. Written postcards narrating where the children have travelled, what they are doing there, what do they like most and what do they miss. |
| People description | Two | Robot sketches, and texts of cartoon characters described. Written descriptions of people following a scheme provided. |
| Ambience description | Four | Drawings of landscapes previously described orally. Brief written descriptions and their drawings. Oral descriptions of landscapes and interior spaces. Oral landscape descriptions and drawings following a provided scheme. Oral and written landscape descriptions, and their drawings. |
| Other genres | Two | Written recipes. Drawings about jobs infrequent among women, and a written motto for each of them. |

In this sense, the aims of our research may be summarised as follows:

1. To describe the multimodal written texts produced by children aged 7 to 8 years old.
2. To analyse the intersemiosis of multimodal children's writing and its relation with the overall development of literacy in young children.
3. To interpret the construction of meaning of multimodal written texts in the school, taking into account the nature of children's literacy experiences acquired out of school.

*2.2. Research Design, Description of Participants, and Data Collection*

The current research was part of a wider research project about literacy development in early childhood from low-income communities in the South of Spain (EDU2017-83967-P) [54]. One of the aims of the project was the examination of the ways in which young children incorporate their own heritage into the classroom, and to draw on them to empower children and let them become the actors in their educational process [55].

The project lasted two years and employed a classroom ethnography design [56]. During the whole project, data were gathered through participant observation, conversations and interviews, and the analysis of documents and children's products, documented through photographs, video and audio recordings (see Table 3).

**Table 3.** Nature, number and duration of the data gathered.

| Data Gathering | Number | Duration |
|---|---|---|
| Children's products | 240 | |
| Photographs | 540 | |
| Sets of notes containing field observation | 47 | |
| Video and audio recordings | 29 | 1020 min. |

The participants in the project were 12 children (six girls and six boys) aged 7 to 8 years old, attending the second year of primary education in a state school in an urban area in Seville, with Spanish as their first language. This paper presents the analysis of two of the written products generated by two girls, Sara and Marina, during the project. They have been selected for their integration of language and visual modes, and for the ways in which these children use their writings to make their own decisions and to integrate their culture and knowledge into their school tasks.

### 2.3. Ethics

The children's and their families' participation in the project was voluntary and followed the ethical requirement for informed consent, that restricts the use of the information obtained for research purposes only, and guarantees the anonymity and confidentiality of the participants. Moreover, this research followed the internal regulations for Social Sciences required by the Ethical Committee for Experimentation of the University of Seville.

### 2.4. Data Analysis

From all the data gathered during the intervention, we chose those oriented to the development of multimodal narrative genres [22,57]. The analysis of the pupils' products (drawn and written texts) was performed drawing on Bearne [58], Moya Guijarro [37], and Painter, Martin, and Unsworth [38]. To this end, a table of indicators containing the three metafunctions [22] of verbal and visual texts, as well as their relations, was designed (see Table 4).

**Table 4.** Analysis of the metafunctions in language and visual texts.

| Metafunction | Language | Visual |
|---|---|---|
| Ideational | Characters: what information is provided, and what is the relation between them. Narrator: who tells the story. Topics, themes or features inspired in traditional tales. Intertexts from the popular culture: television, videogames, internet, etc. Organisation of the processes (verbal and mental actions) and relation between the different parts of the narrative sequence. Use of background detail to create mood and setting. Description of the spatial and temporal frame. | Character's depiction: complete (including head) or metonymic (parts of the body, shadow, silhouette . . . ). Character's appearance and reappearance. Contrast between different characters as a manifestation of their relations. Means for transmitting pathos and affect, such as the images' face expressions, or drawing details (naturalism). Depiction of secondary characters. Processes: actions, verbal, mental (cognition or perception). Inter-event options: What happens between one image and the following. Circumstances: Depiction of the context and its changes. What does the image add to the text? Hypothetical ideational contradiction between text and image. |
| Interpersonal | The view of the narrator, and the means used to implicate the reader in the text (relations of power between the narrator, the characters and the reader). Use of language appealing the reader (references, interpellations, etc.). Language means to connect to the reader's empathy. Use of narrative/informational structure and language to engage and hold the reader's attention. Use of modality in the text. Selection, adaptation, synthesis and shaping of content to suit personal intentions, ideas and opinions. | Focalisation (mediated or unmediated; through contact or observation). Means for depicting social distance. Attitude: involvement and power. Means for transmitting pathos and affect, such as the images' face expressions, or drawing details (naturalism). Ambience: Use of bright, plain or dark colours to attract attention or highlight characters and actions. Effects of the colour on the reader. |
| Textual | Integration and balance of modes for design purposes. Choice of language, punctuation, font, typography, layout and presentational techniques to create effects and clarify meaning. Choice of the mode(s) that will support informative, communicative and expressive purposes. Handling of technical aspects and conventions of different kinds of texts. Marked and unmarked clauses and change of the usual clause structure. Text marks (connectors) between the different parts of the narrative sequence. Choice and use of a variety of sentence structures for specific purposes. Use of concrete or abstract language. Explanation of the choices of mode(s) and expressive devices including words. | Visual organisation and focus. Layout of verbiage and image (integrated or complementary). In complementary layout, which mode is privileged (verbiage or image). Manifestation of meaning (locution or idea) in integrated layout (more likely through bubbles). Framing: detachment of images from verbiage. Relation of the characters and the action with the frame. |

Sources: Adapted from Bearne [58], Moya Guijarro [37], and Painter et al. [38].

Our analysis interprets the contextualised meaning of the children's texts from a semantic-pragmatic perspective. In some occasions, a single verbal clause, or an image, may be interpreted from more than one ideational, interpersonal and textual meanings of the metafunctions. Therefore, our analysis considers the description of the discursive multimodal components of the texts, as well as the intentionality given to them by the children. The interpretation of the children's intentionality is based on the field observation and the conversations between the children and the research team during the writing process of these texts.

## 3. Results

The following results are the analysis of the products generated during one of the sessions, held on 30 October 2019. The session focused on the narrative genre and, owing to the proximity of Halloween, the aim was to create a horror story. The deconstruction stage introduced the main parts of a narration making use of stories told orally. The children incorporated narrations taken from TV and films that served as a starting point to develop their metalanguage awareness. The joint construction stage oriented the narrations towards Halloween. Horror stories were narrated orally, with special attention to characters related to Halloween and well known to children through media, advertising, and the exhibition hosted in the school library during that week. With all these elements in mind, the whole group, guided by the teacher and one of the researchers, designed the beginning of a horror story. Following the format of traditional tales, the story began with "once upon a time" and included a far-away country and a castle supposedly inhabited by a monster. One day, a child named Manuel dared to enter the castle, where he found something unexpected. During the individual construction stage, the children had to finish the story individually. They were asked to provide a resolution for the big question: What did Manuel find in the castle? They were free to change Manuel to any other character they preferred. They could use a free layout or use a six-vignettes model. In either case, it had to be a multimodal text combining verbal and visual modes, and they were free to incorporate any character from popular culture they wished. Hereafter, we present and analyse what Sara and Marina produced.

### 3.1. Analysis of Sara's Tale

Sara's story (see Table 5) is a single multimodal text, where the discourse relies on verbal and visual elements interacting between them. The analysis of this text addresses the metafunctions' meaning-making and, particularly, the way in which transitivity, mood and theme are found in the text.

**Table 5.** Sara's tale transcription.

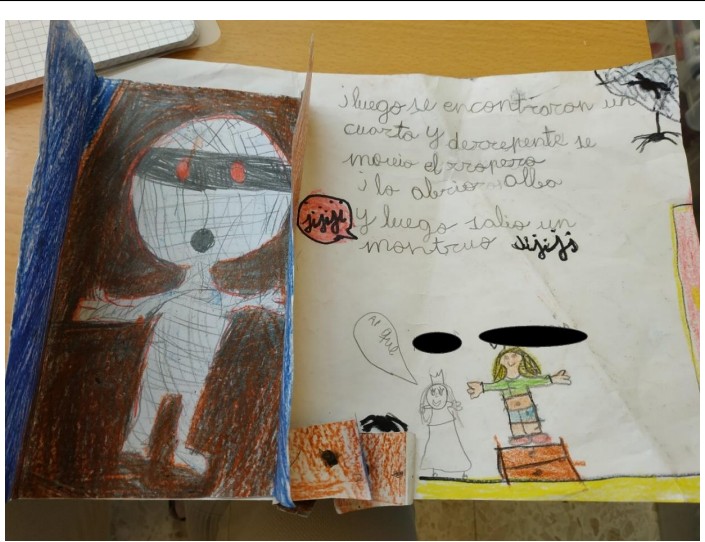

**Table 5.** *Cont.*

| Language transcription | Visual transcription |
| --- | --- |
| "and then they found a room and suddenly it moved, the wardrobe, and she opened it, Alba, and then it came out, a monster". | |
| 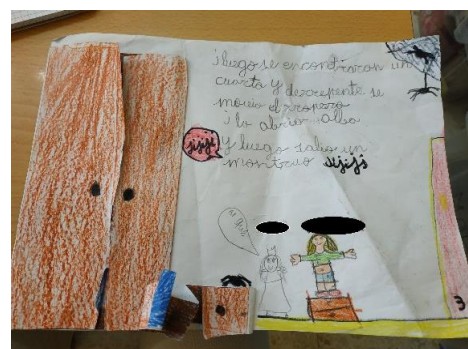 | There is a big wardrobe in the left part of the image. In the bottom there is a desk, and the drawings of two girls, named Marina and Sara, who is standing on a bedside table. There is a spider on the table, and two more spiders with a web in the upper right of the image. A bubble in the verbiage reads "jijiji" on the left, and another one on the right. In the lower part, the girl named Marina says "Oh what", the other girl is named Sara. The doors of the wardrobe may be opened, and inside there is a mummy, much bigger than the two girls. |

### 3.1.1. Ideational Metafunction

The verbal and visual elements which configure the ideational meaning are diverse, as may be seen in Table 4. The verbal factors include the information about the characters and frame, the organisation of the narrative sequence and the processes involved, or the intertexts from other narrative genres and cultures. The visual mode addresses these same topics and the way they are depicted in the image.

In Sara's story, the characters are introduced directly by their names, but they are not described. Sara has substituted the proposed main character, Manuel, with three girls, namely Sara herself and her two friends, Marina and Alba. As this task was introduced in a classroom context, perhaps it was unavoidable for her to want to include her two best friends and classmates. Therefore, her story reflects her previous experience and her perception of her environment. Regarding the visual mode, the characters, including the monster, are depicted in full. Since the story is a single vignette, the characters do not reappear. The monster is the most important character in the story, and it is portrayed as the biggest character. Although it is presented in the verbiage as a monster, it has the precise appearance of an Egyptian mummy. The wardrobe's pop-up links this vignette to the resources present in many children books. One of the characters, Alba, is mentioned in the verbiage but she is not depicted in the image. Two other characters, not mentioned in the text, are drawn. The most relevant is Sara, the author. She is the biggest of the two girls drawn, with her arms wide open and standing on a bedside table. The other is Marina, who is drawn in monochrome pencil and in less detail, even when she is crowned. The two characters are smiling.

Although Sara herself is one of the characters, the story is narrated in the third person, as is common in most of the children's stories. The presence of three characters may also be regarded as an attribute taken from traditional tales such as Goldilocks and the Three Bears. The visual mode incorporates some details inspired by popular culture, such as the conventions of Halloween decorations, like the spiders and webs drawn on the walls and the table.

The processes in this story, included in the transitivity system, are mainly actions, presented as a sequence structured in three stages [28]: the girls find the room (orientation stage), the wardrobe moves (complication stage), Alba opens the wardrobe and the monster appears (resolution stage). There is no verbal description of the background or ambience, and the kind of system used in this metafunction may be included in the category of nuclear transitivity [26]. In contrast with this nuclear transitivity, focused on the process, the image shows no action. The wardrobe opening is not carried out by Alba in the image, since she is not depicted, but by the reader who opens the pop-up. Although the ambience is not completely depicted, some furniture—the wardrobe, a table and a bedside table—reveals the interior of the castle. Thus, the image adds some ambience which is absent from the verbiage.

### 3.1.2. Interpersonal Metafunction

The interpersonal metafunction provides meaning regarding the narrator's view and the interaction between narrator and reader. In the verbal mode, this meaning comes from the mood types, the polarity and the modality systems. In the visual mode, interpersonal meaning is manifested through focalisation, affect, ambience and graduation of the images. Some of the choices made by the children in this metafunction may be also explained as part of the textual metafunction, the difference being the intentionality of these choices.

Regarding the verbal mode, the text offers a neutral positioning of the author. The presence of indicative declarative affirmative clauses (mood types), without negative or non-assertive expressions (polarity system) and with no use of modal verbs, quantifying elements or obligation or willingness expressions (modality system), shows that Sara's discourse does not add any interpersonal meaning in her text. In contrast, the empathy with the reader is shown through the speech bubbles inserted in the text ("jijiji"). Although it is not explained in the verbiage, these bubbles present the monster laughing. They transmit the meaning of Halloween as a merry festivity, in which frightening characters are portrayed usually in a cheerful way. Moreover, one of these bubbles is coloured in red, attracting the attention of the reader and detaching it from the rest of the verbiage. Hence, the empathy in this story is contained in the visual mode rather than in the verbal.

In the drawing, there is no social distance between the characters, who are presented in full, with their arms open, gazing directly at the viewer. There is no interaction between them, who merely appear in front of the viewer. The monster seems to try to scare the children, with his mouth round, apparently howling. Despite that, the characters are smiling, their attitude is friendly and open to the reader, seeking empathy, and they do not seem scared by the monster.

The pop-up wardrobe is one of the most relevant features in the story, and it asks the reader to engage in the action, since it is the reader who opens it and reveals the monster. Its surprising discovery is prepared by Sara in the verbal mode, ("and suddenly"). Through this action's meaning, the reader becomes part of the story, assuming the role of Alba in the verbiage, and therefore becoming an actant.

### 3.1.3. Textual Metafunction

The textual metafunction meaning is provided by the theme and the focalisation of the verbal mode, and by the composition of the visual space. The words in the verbiage refer mainly to the actions ("encontraron" ["found"], se movió ["moved"], "abrió" ["opened"]), which are not depicted in the image. Thus, the temporal sequence of the story mainly relies on the text's verbal mode. The unmarked structure of the clauses highlights the action rather than the actant, since the verb is located before the agent in three of the four clauses. This structure is iterative throughout the verbiage, reminiscent of the iterative structures in oral discourses [59].

The verbiage and the image are integrated in the layout, although there is no interaction between them. With reference to the layout characteristics described by Painter et al. [38], the elements located in the upper part of the image and to the left attract more attention than those in the lower side or to the right. Therefore, the verbiage is privileged over the drawn characters, and more importance is given to the monster and the wardrobe, which are bigger than any other part of the vignette and located on the left side of the page. The rest of the characters are drawn centrally, and minor elements of the ambience, such as the spiders and web to the upper right side of the page, are given less importance. Colours are plain, and they highlight the more brightly-coloured Sara in the lower part of the image. The contrast between the colour of the wardrobe (brown) and that of the monster (white) reinforces the monster's presence, who becomes the dominant feature for the reader/viewer. Meaning is provided in the image through speech bubbles (monster) and thought bubbles ("Ay que" ["Oh what"], by Marina). The vignette is not framed but its upper right limit is integrated in the image, serving as the corner from which the spider's web hangs.

*3.2. Analysis of Marina's Tale*

Marina presented her tale as an illustrated narration divided in six vignettes (see Figure 2 and Table 6). She added some characters and imagined a full multimodal text including different ambiences (the interior and the exterior of the castle). She added some other elements of her own, such as a title and her signature in the back side of the paper.

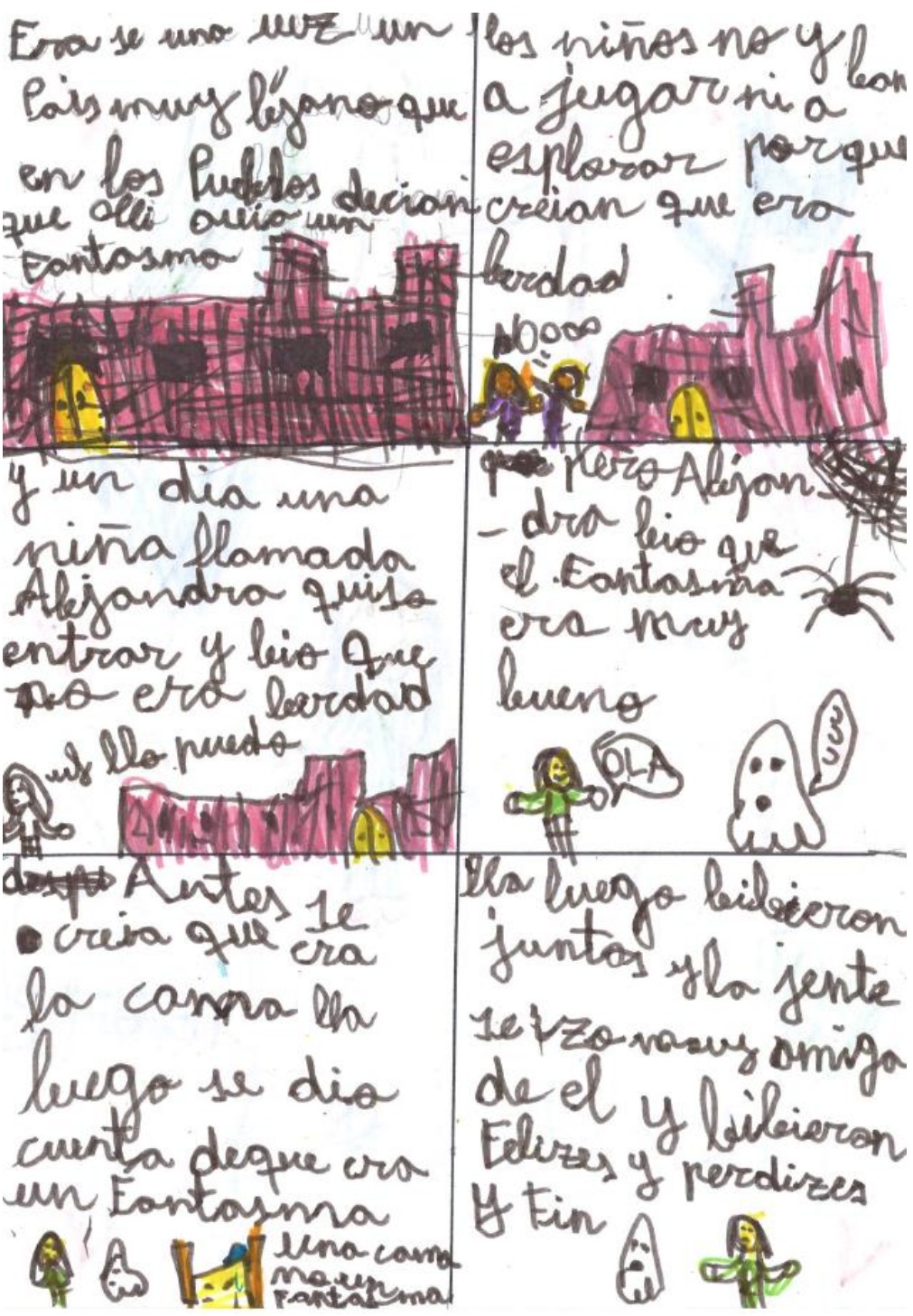

**Figure 2.** Marina's tale.

**Table 6.** Marina's tale transcription.

| | Marina's Story | Language Transcription | Visual Transcription |
|---|---|---|---|
| 1 |  | Once upon a time a faraway land that in the villages they said there was a phantom. | The verbiage is situated in the upper part of the vignette, while the castle's image occupies the lower part. |
| 2 |  | The children didn't go to play or explore because they thought it was true. | The castle remains in the lower part of the vignette, where two characters have appeared, one of them saying: "Noooo". |
| 3 |  | And one day a girl called Alejandra wanted to enter and she saw it wasn't true. | The castle is in in the lower part of the image, and it has appeared a new character, Alejandra, saying: "Uf I can". |
| 4 |  | What But Alejandra saw that the phantom was very good. | The castle has disappeared and in the lower part of the image appear Alejandra saying "Hello" and the phantom saying "Uuu". There is a spider hanging from its web in the upper right corner. |
| 5 |  | Aft Before she thought that it was the bed and then she noticed it was a phantom. | In the lower part of the image there are Alejandra and the phantom, and a bed, with an explanation: "A bed, not a phantom". |
| 6 |  | And then they lived together and the people become close friend of him and they lived happily ever after and end. | The characters of the phantom and Alejandra remain in the lower part of the vignette. |
| 7 |  | And end. Alejandra and the monster. Signature. | The back side of the sheet combines text and the images of the phantom and Alejandra. Other ornaments are present, such as stars and hearts in different bright colours. |

### 3.2.1. Ideational Metafunction

There are two main characters in the story, Alejandra and the phantom. Marina has substituted the proposed main character, Manuel, for Alejandra, the same name as the researcher who proposed the task. In the verbal mode, while Alejandra is not portrayed, the phantom is described as "very good". There are two collective characters, "the people" and "the children", left as generic characters. The story is narrated in the third person, as many children's books are. There is no verbal description of background or the ambience. Visually, the characters are drawn completely, smiling and with their arms wide open. "The children" characters appear in the second vignette as two young people drawn with less detail than the rest of the characters, thus depicting their secondary role and anonymity. Alejandra appears in the third vignette, and she reappears in the rest of the vignettes, accompanied by the phantom. They always have the same appearance (clothes, hair, smile, size), with no development or change in their relations: they always appear side by side, without direct physical or eye contact. There is balance between Alejandra and the phantom, since they are the same size and treated equally by Marina. This balance reinforces the idea of the "good" phantom.

The story incorporates several elements inspired by traditional tales. The beginning ("Erase una vez un país muy lejano" ["Once upon a time a faraway land"]) was provided by the researcher, but the ending ("y vivieron felices y perdices y fin" ["they lived happily ever after and end"]) is the child's own product. The reverse side of the page also contains a closing formula ("y fin" ["and end"]) as well as the story's title ("Alejandra y el monstruo" ["Alejandra and the monster"]), Marina's signature and the drawings of the phantom and Alejandra. The visual mode also contains a reference to popular culture, when the phantom is portrayed in a way very similar to the character of the Pac-Man videogame, thus linking the visual mode to popular culture and Marina's previous experience.

The transitivity system focuses on the processes, being these actions ("ya luego vivieron juntos" ["then they lived together"]), and mental processes, both cognitive ("se dio cuenta de que era un fantasma" ["she noticed it was a phantom"]) and perceptive ("vio que el fantasma era muy bueno" ["saw that the phantom was very good"]). The story follows the narration steps as described by Martin and Rose [28] as Orientation (vignette 1), Complication (vignettes 2 and 3) and Resolution (vignettes 4, 5 and 6), with a Coda on the reverse of the page. Images do not contradict the transitivity meaning of text. On the contrary, they rather reinforce it and provide an illustration to the actions, in a static way. One of the very few movements depicted is the children trembling in front of the castle (second vignette). The background changes from the outside to the interior of the castle. In the first three vignettes, the castle is depicted in a similar way—towers, the colour of the door and walls, the number of windows. In the fourth vignette, the narration moves inside the castle, and this interior is represented by a spider in the upper right corner and the absence of the castle's image. In the fifth vignette, one of the writer's aims is to explain a confusion; Alejandra is in the room, and she thinks she is in front of a bed, but actually she is in front of the phantom. Therefore, the bed is drawn in the vignette, thus marking this difference and the character's confusion.

### 3.2.2. Interpersonal Metafunction

The meaning of the interpersonal metafunction in Marina's tale is displayed as a network of references appealing to the reader. One of the most relevant is the inclusion of one of the researchers as the main character. The story was generated as a school task proposed by a member of the research team, Alejandra, and locating her as the main character may be interpreted as a way to create a direct empathy with the reader.

In the verbal mode, the interpersonal meaning may be found in the mood types, and the polarity and modality systems. Regarding the mood types, all the clauses are indicative declarative affirmative. In the polarity system, all of the clauses are affirmative except "los niños no iban a jugar ni a explorar porque creían que era verdad" ("The children didn't go to play or explore because they thought it was true"), which may be described as a negative/non-assertive expression. Marina's offers a greater number of resources in the modality system, where she makes use of the verbs of belief ("Antes se

creía que era la cama, ya luego se dio cuenta de que era un fantasma" ["Before she thought that it was the bed and then she noticed it was a phantom"]; "Los niños no iban a jugar ni a explorar porque creían que era verdad" ["The children didn't go to play or explore because they thought it was true"]). Thus, the modality system is used to create a distance between reality and the beliefs of the people about the castle ("en los pueblos decían" ["in the villages they said"], "porque creían que era verdad" ["because they thought it was true"]). In that way, Marina shows that she does not share the people's and children's opinion about the castle, and mood system acquires a relevant position in the interpersonal meaning.

For its part, the focus in the images is neutral, centred on the characters. They want to be friendly and are depicted as communicating with the reader (they smile). The castle is coloured in pink, contrasting with the white background. The door, coloured in yellow, is clearly visible, attracting the viewer's attention towards it and foretelling what is going to happen—someone entering the castle. Marina uses the speech and thought bubbles to create empathy with the reader ("Uf yo puedo" ["Uf I can"]), but also to explain directly to the reader some parts of the narration that may be confusing, as happens in the fifth vignette where Marina uses the image to clarify Alejandra's confusion ("una cama no un fantasma" ["a bed, not a phantom"]). Therefore, interpersonal meaning in Marina's story is present mainly on the verbal mode, in which mood expresses subtle nuances of meaning, such as the author's own opinion about the "people's" belief.

### 3.2.3. Textual Metafunction

Marina relies on the verbal mode to direct the story and, through the vignettes' visual layout, relevance is given to the verbiage, located always in their upper part. The images and the verbiage are complementary, that is, they appear in different spaces in the layout [38]. The vignettes are separated by frames, and the characters do not interact with the frame. However, in the fourth vignette, the frame is treated as an architectural element of the castle's interior, as the spider's web is hanging from it, as it would be from a corner in the castle. When the castle appears, it becomes the most important part of the drawing, although in the third vignette, when Alejandra turns up for the first time, the building is smaller than in the previous vignettes. In such a way, Marina also reorients the reader's attention towards the characters.

The declarative clauses in her discourse possess an informative structure, unmarked in Spanish. The temporal order of the processes is delineated through a variety of connectors ("pero" ["but"], "antes" ["before"], "luego" ["then"], etc.) with a cohesive function within the text. In some vignettes, Marina doubts the proper connector, as shown in the crossed-out beginnings in the fourth and fifth vignettes. All the action verbs are in past tense, also contributing to this temporal cohesion, and pointing to the literary sources of the text, given that the past tense is the most commonly used in traditional and children's tales. In her clauses, Marina shows a range of resources, making use of the indirect style ("decían que allí habían un fantasma" ["they said there was a phantom"]) and a diversity of subordinate clauses ("porque creían" ["because they thought"], "vio que" ["she saw that"], etc.). Direct style, for its part, is found in the speech bubbles, as found in the encounter between Alejandra and the monster. Some of these structures are clearly inspired by the formulae used in the written texts (e.g., "una niña llamada Alejandra" ["a girl called Alejandra"]).

Regarding the visual mode, the display of the different elements in the images complement the actions of the verbiage. At the beginning, when the narrator introduces the country and the castle, only the castle appears, occupying the full space of the image. As the characters begin to appear, the castle becomes smaller. Alejandra is always on the left side of the image, as a relevant character in the story and more active than the phantom. Meaning appears in bubbles in the images. In them, locution meaning (fourth vignette, when Alejandra and the phantom meet) and ideas (third vignette, Alejandra encourages herself, "Uf yo puedo" ["Uf I can"]) are found. In the fifth vignette, the image is explained by the narrator in a bubble ("una cama no un fantasma" ["A bed, not a phantom"]), as an attempt to further clarify Alejandra's confusion.

## 4. Discussion

The analysis of the artefacts produced by Sara and Marina has showed the complexity of young children's writings, and the intermodality of their written texts. In them, the relationship between the verbal and visual modes jointly builds the texts' global meaning. The complexities of multimodal texts demand a separate analysis of their modes, and a subsequent study of their intermodal relation [38].

Language creates reality through categories, generating a transitivity system which determines the participants who take part in the action, the circumstances that surround them, and the processes [23]. The visual images represent reality making use of other semiotic codes to depict locations, characters, the author's and reader's points of view, and contributing to the textual cohesion [60]. In this sense, O'Halloran [44] poses some meanings suitable to be built into a bimodal text. The concept of intersemiosis enables an approach to the convergence of meanings (co-contextualizing relations) or their divergence (re-contextualizing relations), both in the verbal and the visual mode within the same text. The elements comprising the multimodal textual coherence contribute to the configuration of diverse points of view within the frame of ideational, interpersonal and textual meanings.

The case of Sara is an example of intersemiotic relations within a complementary meaning system [38]. This system articulates the meanings provided by image and language, where the verbal mode brings some meanings lacking in the visual mode, and vice versa, and both contribute to build the whole of Sara's story. Ultimately, the meanings across the ideational, interpersonal and textual metafunctions are built through these intersemiotic relations in Sara's story. Firstly, both the visual and verbal modes provide the ideational meaning, manifesting the way in which the child interprets the world. For example, the verbal mode incorporates two characters—Alba and the monster—and the visual mode presents three: the monster, Sara and Marina. Secondly, the interpersonal meaning of Sara's story relies on the visual mode, which manifests empathy and interaction with the reader—the monster's speech bubbles, or the wardrobe doors' pop-up—whereas only a slight interpellation to the reader is found in the verbal mode ("y de repente" ["and suddenly"]). Finally, the meaning of the textual metafunction concentrates on the theme system, through the unmarked structure of the clauses across the text, while the layout of the visual mode focuses on the monster and its placement in the wardrobe. Therefore, the meanings provided by the verbal and the visual modes are complementary in all the three metafunctions.

For its part, the intersemiotic relation of the verbal and visual modes in Marina's story may be described as a symmetrical interaction [61]. Throughout Marina's story, the same information is repeated across two different semiotic codes. In practice, this iteration implies that one of the modes leads the story, and the other provides additional meanings, based on the connotations of the bimodal text's reading. On the one hand, the ideational meaning in Marina's story draws on the verbal mode for the display of the narration's transitivity, and the cognitive and perceptive mental processes [38]. On the other hand, the visual mode illustrates the manifestation of the characters, the background, and the process developed in the verbiage. The interpersonal meaning in the verbal mode expressed in the modality system marks the author's positioning towards some characters' opinions (e.g., "en los pueblos decían" ["in the villages they said"], "porque creían que era verdad" ["because they thought it was true"]). At the same time, the visual mode incorporates the empathy with the reader through the thought and speech bubbles (e.g., "Uf yo puedo" ["Uf I can"], "Noooo"). Finally, the textual meaning of the verbal mode takes precedence over the visual which is especially noticeable in the visual layout of the vignettes.

In sum, the two cases analysed have allowed the researchers to verify differences in the multimodal writing in the first years of schooling. The first case—Sara's story—requires the visual mode to complete the information provided through the verbal mode, and to build an efficient narrative in a global sense [62]. On the other hand, Marina's story shows independence in the use of the verbal mode compared to the visual. Likewise, their texts display the diverse intertexts employed in the construction of their stories [45]. Sara, for instance, presents a story based on children's tales with pop-up elements, as shown in the wardrobe to be opened by the reader. Marina's story incorporates precise verbal

marks coming from traditional tales and books where the visual mode is redundant compared to the verbal mode. The intersemiotic relations present in Sara's and Marina's writings show their different approaches to reading and writing. These approaches could be conditioned by their distinct literacy expertise, and would explain why Sara chose a complementary meaning system. However, the accurate use of the modes in Sara's and Marina's writing shows their different literacy sources (pop-up books or traditional children tales), characterised by different intersemiotic relations. Thus, both children incorporate their literacy knowledge in their multimodal writing. Furthermore, our analysis shows how Sara and Marina take the particular subgenres of multimodal children's books as models, applying their literacy knowledge acquired at home. Therefore, a genre-based pedagogy in the first years of school should prioritise multimodal genres, benefiting from children's literacy knowledge acquired in and out of school.

The differences between the two cases point to the assessment of literacy development, showing the need to analyse the relationship between the verbal and the visual modes in the first stages of writing [63]. The intermodality of the children's writings requires greater attention by teachers in order to understand the complexities of the multimodal texts produced in the primary school years [39]. If we assume the multimodal nature of many of the texts surrounding children from an early age, and aim to incorporate them in the classroom, teachers have to design tools enabling them to assess children's writing from a multimodal point of view. The different intersemiotic relations present in children's multimodal products, as shown in our analysis, demand the evaluation of all the modes present in them, and not only the verbal mode. Therefore, assessment of children's writing should incorporate indicators oriented to multimodality. The items used in our analysis, contained in Table 4, may be the source for these assessment indicators, and could be incorporated by teachers in their design of the teaching-learning process.

The written products analysed reflect the ways in which the children incorporate their social context, heritage and knowledge into the classroom. The children were given freedom to choose the layout, characters, settings, plot, etc., of their stories. They made use of this freedom by changing the main character, creating some secondary characters, choosing the story's layout and plot, and following narrative conventions and patterns. The use of certain hypertexts (comic, pop-up) and iconography (the Pac-Man phantom, the spiders) shows the way in which children incorporate their knowledge acquired outside the school. This writing task became, in this way, a space for participation and children's interaction with the school. The inclusion of one of the researchers as a character in one of the stories, for instance, strengthens the intergenerational dimension of participation. Ultimately, the cultural relations established between the children's context and the school through this task, and the value assigned to the children's voice, may be considered as an example of the contribution of literacy education in schools to the development of social sustainability.

Our research has explored the multimodal texts of children aged 7 to 8 years old, analysing the ideational, interpersonal and textual metafunctions' meanings, through the intersemiosis between the verbal and visual modes [64]. This in-depth analysis has permitted the researchers to interpret the construction of meaning in children's multimodal texts, taking into account their complexities and diversity [65]. Both Sara and Marina make use of their multimodal resources in a consistent way, developing clear and well-defined intersemiotic relations. Their mastery highlights the relevance of the literacy events taking place out of school, and their complementarity with those incorporated to educational settings. The exclusion of multimodal genres from the classroom would imply the neglect of a wide range of semiotic resources already known by children. As a result, the analyses underline the need to incorporate a genre-based pedagogy to the school curriculum, an incorporation that would demand the teachers' awareness of the assessment of multimodality within children's literacy development. In this sense, our research challenges any educational system which considers literacy as the acquisition of reading and writing skills disconnected from the children's environment. From our perspective, literacy dialogues with the children's context and incorporates it into the school context. As far as this process is able to integrate the children's knowledge of the world and their

wisdom, acquired outside the classroom, multimodal writing may become a way to develop key factors to social sustainability development.

**Author Contributions:** A.P.-C. and F.G.-S. conceived, designed, and wrote together this paper. All authors have read and agreed to the published version of the manuscript.

**Funding:** This research has been carried out within the framework of I+D Project: Literacy as a social practice in pre-school and primary education (5–7 years old): Research and Intervention design on children at risk of social exclusion in urban contexts (EDU2017-83967-P), funded by Spanish State Research Agency, Ministry of Economy, Industry and Competitiveness and the European Regional Development Fund (ERDF).

**Conflicts of Interest:** The authors declare no conflict of interest.

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
