# Peer review of "Developing Multimodal Narrative Genres in Childhood: An Analysis of Pupils’ Written Texts Based on Systemic Functional Linguistics Theory"

_education, doi:10.3390/educsci10110342_

Round 1

Reviewer 2 Report

The research is valid and relevant, but it should be presented within a more clear and delimited approach. As it is now, it is, in some aspects, vague and confusing. In other words, the particular purpose should be clear-cut and straightforward. Multimodality is a rather vast world, and, since the SFL framework provides such relevant tools for a scientific look, it will be helpful to read a grounded and systematic analysis.

The paper has all the potential and the resources, but it is somehow difficult to understand, specially regarding the following interrogations:

  • The particular aims (expressed from lines 191-94) are made clear. But what is the relation between them and a general concern with the development of literacy, as well as with all the background presented in the beginning of the article? A more explicit reasoning is needed in this sense.

  • What are the particular meanings that the current analysis brings? What is the specific relevance of noting that there are two different inter semiotic relations for the analysis of writing? A more explicit reasoning is needed in this sense.

  • Is there any substantial evidence to say that “the analyses underline the need to incorporate a genre-based pedagogy…“ (lines 528-30)? What this, actually, part of the research goals? A more explicit reasoning is needed in this sense.

From a global point of view, a general point that needs clarification: does this research makes distinction between the children’s development and the teaching/learning? 

It is rather vague and somehow confusing. What kind of perspective will be put forward: a developmental one? The research that is presented gives a view of the complexity involved, but does not develop a strict developmental approach. 

A statement may need rephrasing: “The Systemic Functional Linguistics offers a framework for the development of children’s literacy in a multimodal way.“ (part of the abstract)

The learning methods from which the shift is to be applied are not clear. (Lines 55-56)

The theoretical summary presented in 1.1. includes general statements that have may be interpreted as too simplistic and erroneous. Given the detailed comments on the children’s products, it is necessary to present a global view on the SFL framework. But special attention should be addressed to the overview on the relation between register and genre (lines 69-74)

Although the genres themselves are not the focus of the research, they play a relevant role in the work done in the classroom (since it is repeatedly stated that it is genre-based pedagogy). Therefore, I would suggest:

  1. A clearer view on the itens listed under ‘genres’ and on their arrangement. On the one hand, is the item ‘narrative’ taken as one genre? On the other hand, are “people description” and “ambient description” different genres?
  2. Some thoughts could be provided on the relation between the genre label chosen and the list of products considered.

It is not sufficiently explained how the research points to the need of a genre-based pedagogy or of the awareness of the assessment of multimodality.

I suggest a more direct and concise reflection about the data analysis.

The results are complex, are rich and are particularly well described. Lacking is their meaning. And the way this relates to teaching and assessing.

One reference is missing on line 108.

The provided hyperlink in line 559 is not retrievable.

Reviewer 3 Report

The research is well conducted and explained, as well as the discussions of the results. But there is a variable missing from the onset. The researchers do not mention that the study was conducted with children living in Spain, and do not specify what is their L1. The reader infers this when the name of the University of Seville is mentioned on page 6, and because of the written examples taken from the two children production artifacts. This is an important piece of information because the reader would like to know if the results obtained in this research would significantly vary depending on the L1 or several L1 the children speak.
